# Ecotourism Disturbance on an Endemic Endangered Primate in the Huangshan Man and the Biosphere Reserve of China: A Way to Move Forward

**DOI:** 10.3390/biology11071042

**Published:** 2022-07-11

**Authors:** Wen-Bo Li, Pei-Pei Yang, Dong-Po Xia, Michael A. Huffman, Ming Li, Jin-Hua Li

**Affiliations:** 1Key Laboratory of Animal Ecology and Conservation Biology, Institute of Zoology, Chinese Academy of Sciences, Beijing 100101, China; lwb@ioz.ac.cn; 2International Collaborative Research Center for Huangshan Biodiversity and Tibetan Macaque Behavioral Ecology, School of Resources and Environmental Engineering, Anhui University, No. 111, Jiulong Road, Hefei 230601, China; ahppyang@126.com; 3International Collaborative Research Center for Huangshan Biodiversity and Tibetan Macaque Behavioral Ecology, School of Life Sciences, Anhui University, No. 111, Jiulong Road, Hefei 230601, China; dpxia@163.com; 4Wildlife Research Center, Inuyama Campus, Kyoto University, Kyoto 606-8501, Japan; huffman.michael.8n@kyoto-u.ac.jp; 5School of Life Sciences, Hefei Normal University, No. 1688, Lianhua Road, Hefei 230601, China

**Keywords:** Man and the Biosphere reserve, sustainable development, ecotourism, Tibetan macaque, species distribution modeling, MaxEnt

## Abstract

**Simple Summary:**

How to realize the sustainability of economic development and animal protection is a significant problem faced by Man and the Biosphere reserves. Although there are many theoretical frameworks, there is still a lack of supportive ecological evidence. This study analyzed aspects of the local human population, economic growth, number of tourists, and ticket income data of Huangshan Man and Biosphere Reserve (HMBR) as well as population and distribution changes in the flagship species (Tibetan macaque) in HMBR over a 30 year period. We found that after 30 years of implementing a sustainable development strategy in HMBR, the local economy and the population of Tibetan macaques have increased simultaneously. With economic growth, more funds for protection have been invested, improving the local environment significantly and expanding the existing distribution of the Tibetan macaque population. This study provides strong evidence for the sustainable development of Man and Biosphere reserves. We propose that economic and wildlife population growth and distribution area measures constitute a critical standard for the evaluation of sustainable development.

**Abstract:**

The primary purpose of the Man and the Biosphere Program is the sustainable development of both the economy and nature conservation activities. Although the effectiveness of eco-tourism to reach this goal has been proposed, due to the lack of long-term monitoring data and a model species, there has been no obvious mechanism to evaluate the effectiveness of this policy. This study explored the effectiveness of the sustainable development policy of HMBR based on 30 years data of monitoring the Tibetan macaque, local human population, visitors, and annual ecotourism income in Huangshan by estimating species habitat suitability and the impact of ecotourism. The results showed increases in the income for the local human population, the number of visitors, and annual eco-tourism. Simultaneously, the reserve’s Tibetan macaque population size and suitable habitat areas increased. The macaques expanded their habitat to the low-altitude buffer zone (400–800 m), an area with lower eco-tourism disturbance. Scenic spots had a significant negative impact on habitat suitability (the substantially increased contributions of scenic spots from 0.71% to 32.88%). Our results and methods provide a suitable evaluation framework for monitoring the sustainable development and effectiveness of eco-tourism and wildlife conservation in Man and the Biosphere reserves.

## 1. Introduction

UNESCO first proposed the Man and the Biosphere (MAB) program in 1971. It conceptualized locations for reserves to serve as important ecological areas that play an essential role in building a harmonious relationship between humans and nature. Its goal is to allow for both sustainable economic development and the conservation of biological diversity [1,2,3]. There are currently 727 MAB reserves in 131 countries, including 22 transboundary sites, around the world, located in biodiversity hotspots (https://en.unesco.org/biosphere/wnbr) (accessed on 1 June 2022). Most of these reserves face the dual challenges of sustainable economic development and biodiversity conservation, especially in mountainous areas, where the economy is dependent on natural resources for its growth and development. Activities such as logging, farming, and grazing aggravate conservation efforts and lead to the destruction of the natural environment [4]. Rapid economic development comes with severe environmental deterioration and immediate biodiversity reduction [4,5]. With enhanced public awareness of the need for better protection, more of these reserves have begun to find a sustainable development approach through eco-tourism [5]. 

Eco-tourism emphasizes that all links involved in tourism must be sustainable, ecological, and fulfill all basic environmental responsibilities. The MAB implementing ecotourism must rely on their scientific, academic, historical, societal, cultural, and esthetic values and rich biologically diverse natural resources to responsibly develop the tourist industry [6,7]. The apparent advantages of eco-tourism are said to be: (a) contributing to sustainable management and conservation of natural resources; (b) providing alternative means of livelihood for residents to change their view that a nature reserve restricts economic development; (c) offering more economic support for biodiversity conservation and management [7,8,9,10]. However, eco-tourism practices may also have some negative impacts. For example, these activities may change the habitat characteristics of animals and plants. Scenic spot-related infrastructure, such as road and building construction activities, may affect land cover characteristics and ultimately change the distribution of animals [9,11,12]. Additionally, rapid, and irrational forest restoration policies can create “empty forests” [11,12]. Because of the lack of valuable long-term monitoring data, strong predictive models, suitable indicator species, and a sound evaluation system, it is complicated to adequately demonstrate the effectiveness of sustainable development policies in such areas.

An important role of conservation biology is preventing biodiversity loss and destroying important biological communities [13]. In this ever-changing world, conservation biology can play a role in evaluating sustainable economic development [13]. A well-known case study is the giant panda (*Ailuropoda melanoleuca*) of China. Not only was the conservation of the species successfully realized, but those efforts continue to drive the development of the local economy, making conservation no longer an obstacle but a means to economic growth [14,15]. Animals are an essential part of the natural ecosystem. Research and preservation of animal habitats, distribution, and demography are currently some of the more critical activities of conservation biology [13]. In the background of eco-tourism in most MAB reserves are the concerns about the disturbances of eco-tourism on animal populations and the problem of evaluating the feasibility of sustainable development [10]. Previous studies in these areas have mostly paid attention to economic benefits rather than the state of biodiversity and the population dynamics of wild animals affected by eco-tourism [7]. 

At present, only a few long-term studies are looking at the effects of eco-tourism on habitat quality, distribution, and demography of a single species. Additionally, few practical assessments of sustainable development policy have been made [16,17,18]. Long-term research on a single species is expected to help assess the effectiveness of sustainable development policies on conservation. It is not only particular species but also the environment in general that supports many other species [19]. Many tourists, sightseeing routes, and scenic spots generated by long-term eco-tourism activities are known to change the heterogeneity of a landscape in protected areas and disturb the movement of wild animals [9,19]. Multiple studies have shown that the impact of tourists walking activities can also change the spatial distribution of soil structure and vegetation [20]. High tourist activity can substantially increase the availability of human food to endangered animal species, leading to enhanced reproductive capacity and unnaturally large group size [21,22]. The presence of tourists can also change the travel distance of a species [23,24]. Human disturbances such as the construction of tourist routes and scenic spots may have substantial adverse effects on animal distribution patterns, reducing their habitats’ suitability [25,26]. This can lead to drastically altered population structure, increased aggression due to competition for resources, and even the eventual extinction of a species [24]. The long-term negative and potentially positive impacts of eco-tourism on animals need further investigation. 

Primates are good models for this, as they serve essential ecosystem functions as predator, prey, seed dispersers, herbivores, and ecosystem engineers. Nearly 60% of all primate species have been threatened with extinction due to habitat loss and degradation caused by climate change and human activities [27]. Many studies have confirmed that the loss or reduction in primate populations may lead to a cascade reaction. They ultimately result in long-term negative impacts on plant diversity, forest regeneration, and ecosystem resilience [28,29,30]. Because many primate field studies are short-term and often focused on one species, it is not always clear how the changes in landscape attributes affect them in the long-term eco-tourism process [31]. Vital information such as the impact of anthropogenic activity, habitat quality, and forest fragmentation on primate habitat and population dynamics is relatively scarce [32,33]. As most primates are forest-dependent, they will inevitably face many problems caused by long-term coexistence with humans [34,35]. For example, increased encounters with tourists, a decline in forest substrate resource quality, and reduced connectivity of foraging patches can lead to changes in foraging behavior, increased prevalence of disease, food competition, and stress [36,37]. All these factors pose a threat not only to primate conservation but also to regional economic development.

Huangshan, a mountainous area of China, possesses unique geographical diversity, rich ecological resources, and ancient cultural heritage. It became a World Natural and Cultural Heritage Site in 1990 and a UNESCO Global Geopark in 2004. It has provided a valuable foundation for developing eco-tourism. From 1990 to 2004, the annual number of tourists significantly increased from 669,800 in 1990 to 1,601,800 in 2004, an increase of nearly 2.5 times. In 2018, Huangshan officially joined the UNESCO World Biosphere Reserve Network, becoming the 34th MAB reserve in China [38]. Due to its unique geography, culture, and rich biodiversity, the annual number of national and international tourists exceeded 3 million in 2019. It stimulated much socio-economic and sustainable development activity. In addition, it is also a global biodiversity hotspot and one of China’s 32 inland biodiversity priority protection areas [39]. The area contains 2385 plants and 417 vertebrate species. It represents 0.044% of China’s total land area but contains 6.92% of the plant species and 9.55% of animal species in China [40]. The HMAB’s role in conserving biodiversity has become the center of attention. However, in recent years, with the increase in ecotourism, more roads, scenic spots, and buildings are constantly being built and renovated. Whether these have caused damage to the ecosystem of HMAB has not been studied. In addition, there is a lack of practical data to support whether the food refuse left by tourists or directly fed to animals impacts their populations and distribution. Finally, the economic growth dividends brought by eco-tourism have increased the income of local farmers and reduced their dependence on farmland. Most of the cultivated land has now become forest land. Will this reforestation have a positive impact on the distribution and population of animals? Achieving the sustainable development of nature and tourism is essential for preserving the biological diversity of Huangshan. This is an urgent task that requires scientific assessment and management. 

An important keystone species in HMAB is the Tibetan macaque (*Macaca thibetana*). The macaques are classified as Near Threatened and is endemic to east-central and south China [40,41]. They are widely distributed across 13 provinces in China [40,42]. The HMAB is an essential habitat for the species in its northwestern range in China. It is also an ideal species and habitat for understanding the adaptation and evolution of primates. In the early stages of research here, the altitudinal distribution of macaques was 600–1800m, ranging over the entire area. The population was estimated to be 294 individuals across nine groups. They feed mainly on leaves, flowers, fruits, and seeds, adapting flexibly to seasonal food availability. In addition, because of the loss of large mammal predators such as the black bear (*Ursus thibetanus*) and clouded leopard (*Neofelis nebulosa*), the most significant pressure on macaques now is the decline in habitat quality caused by economic development from increased tourism and other human activities [43]. There was excessive hunting activity carried out in this area until the late 1980s [44], but this has now been stopped. Since 1983, the behavioral ecology and population dynamics of the Yulin Keng I group (YAI) of wild Tibetan macaques has been continuously studied in the HMAB. In recent years, it has been found that the Tibetan macaque population (around YAI) has increased significantly, and human–monkey conflicts often occur. Therefore, there is an urgent need to carry out special investigations and research on the population and distribution of Tibetan macaques.

Given the importance and urgency of the issues described above, data and available information about the development of the local tourism economy was used to evaluate the effectiveness of a long-term sustainable development strategy from the perspective of biological conservation. After more than 30 years of afforestation activity in Huangshan, the promulgation of relevant laws and regulations, and strict environmental assessment standards, we predict the following: (1) the population of Tibetan macaque has been partially restored; (2) with the development of eco-tourism, the macaques show a tendency to shift their range away from the areas of severe human disturbance to areas of minor disturbance; (3) with the implementation of sustainable development and an eco-tourism strategy, the size of the suitable habitat area of macaques has increased. 

## 2. Materials and Methods

### 2.1. Study Site

The study was conducted at Huangshan, in southern Anhui province, Eastern China (E 118°01′–118°17′ and N 30°01′–30°18′) (Figure 1). The area ranges from 300 to 1864 m.a.s.l. with an area of 160.6 km^2^ and is characterized by steep slopes and narrow valleys. The rugged granite cliffs in the study area contain nine scenic tourist spots: Wenquan, Yuping, Beihai, Songgu, Yungu, Yanghu, Fuxi, Fugu, and Diaoqiao. The site is located in the marginal zone of the subtropics with high mountains, the highest being Lianhua Peak at 1864 m.a.s.l. The climate changes with increasing elevation, more cloud and fog cover, higher humidity, and more precipitation, forming a monsoon climate particular to the mountain area. The annual rainfall is 2369.3 mm, and it rains 180.6 days a year, mainly from April to June. The yearly average temperature is 7.9 °C, the maximum temperature in summer is 27 °C, and the minimum temperature in winter is −22 °C [38]. The abundant, diverse, and zonal distribution of vegetation supports an ideal habitat for Tibetan macaques and other animal species. Tibetan macaques, for the most part, avoid humans and live in the dense forests in the mountains and cliffs. They prefer subtropical deciduous broad-leaved forests (600–1100 m), and evergreen broad-leaved forests (1100–1400 m). They also occasionally visit the dwarf forest and shrubs (1400–1650 m) and alpine meadows (1650–1840 m). In spring, they move down to low-altitude areas for bamboo shoots (500–600 m) (Wang and Xiong, 1989) (Figure 1). They rarely go to human settlements to forage. Because of their omnivorous diet, monkeys in areas with a long-term tourism presence have become overly accustomed to humans. They are known to attack them over food.

### 2.2. Sampling Methods

(1)Primate population distribution and density survey methods

The historical distribution and population data of Tibetan macaque surveys and census methods were based on work conducted by Wada et al. (1987, the first survey) [44]. The long-term population data for the Yulinkeng I group (a long-term field study population of Tibetan macaques in the current HMBR) was collected over a period of more than 30 years by *the International Collaborative Research Center for Huangshan Biodiversity and Tibetan Macaque Behavioral Ecology* (http://hsmonkey.ahu.edu.cn/, accessed on 1 June 2022). The current distribution and population data were based on surveys and censuses conducted in the period March 2020 –March 2021 (the second survey). The first survey was conducted mainly using the transect and artificial tracking methods [44]. The study areas of the first (1973–1987) and second (2020–2021) surveys overlapped entirely. The monkeys mainly inhabit forests and seldom have contact with humans [43]. The methods used were line transect sampling and camera traps in the second survey [45,46,47]. We divided the study area into 59 grids (2 km × 2 km) in ArcGIS (Version 10.4.1). Considering the topography, vegetation characteristics, and grids’ distribution, 59 line transects (2 km long), were selected, running north and south, east and west (Figure 1). Each transect line was set up with one infrared camera, which was periodically moved to different locations (see Figure 2).

(2)Long-term macaque population dynamics

Due to the lack of extensive long-term monitoring of macaque populations across the study area, we analyze the population demography records of just one long-term study group, the Yulinkeng population in the southern part of the study area (Li et al., 2020). Researchers have recorded this group’s population size and composition since 1986 [40]. The group fissioned four times, and each time, the dominant group always remained in the vicinity of Fuxi village, while the subordinate group dispersed elsewhere. All the individuals in the study group could be individually recognized by facial features, hair color, age, and body size. Each individual is assigned a unique name and traced throughout its life in the group.

(3)Human population size, number of tourists, eco-tourism income, and GDP

We could not directly obtain the population size and economic details of the human residents in the study area for logistical reasons. As a proxy for this data, we accessed the *Huangshan City Statistical Yearbook* for the population and GDP data of Huangshan District from 1985 to 2020. To analyze the development of the area’s eco-tourism industry, we obtained information on the annual tourist arrival statistics and ticket receipts from the *Huangshan Scenic Area Management Committee* from 1980 to 2019. Due to the lack of some data records, there are certain differences in the years of recorded data. However, we only pay attention to the changes in GDP and tourism ticket revenue from 1985 to 2019.

### 2.3. Data Analysis and Modeling

(1)Overall macaque population distribution and density

Based on the results of infrared camera traps, monkey occurrence data was collected. Line transects were employed to estimate the population distribution and density of Tibetan macaques in the area. When we found direct or indirect evidence of monkeys (feces, feeding traces, footprints, etc.), we took photos and recorded their locations with GPS. The population’s age–sex class was classified according to Li et al. (2020) The age–sex classes were determined based on differences in body size, hair color, facial color, reproductive organs, and behavior [40]. According to Wang and Xiong (1989), the distribution range and population size was determined by the average home area size (6 km^2^) and daily range distance (1–4 km) [43].

(2)Population dynamics

From the long-term data of the Yulingkeng group, the following parameters were calculated.

(1)The annual net population growth rate is calculated as follows [48]: R = [(N_t+1_/N_t_ − 1) × 100%.

t is the number of years, N_t_ is the group size at the beginning of the year (January), and N_t+1_ is the group size at the end of the year (December).

(2)Annual birth rate: b = I_t_/F_t_ (Subcommittee on Conservation of Population, 1981). I_t_ is the number of births in year t, F_t_ is the number of adult females.(3)Annual mortality is the number of deaths within one year divided by group size at the beginning of that year.(4)The annual immigration and emigration rates were calculated as the number of immigrating and emigrating individuals divided by group size at the beginning of the year.(5)The operational sex ratio was calculated as the total number of adult males/ the total number of adult females in the group that year.

(3)Evaluation of habitat suitability spatiotemporal dynamics

Maximum Entropy (MaxEnt) modeling was used to evaluate the spatiotemporal distribution dynamics of Tibetan macaques. This method allowed a larger-scale prediction of potential macaque distribution areas beyond the sampling areas with high predictive accuracy [15,34,37,49,50,51]. The selection of prediction variables is based on previous studies of primate habitat suitability [37,39,40].

#### 2.3.1. Environmental Variables

To project and map the habitat suitability of Tibetan macaques, wide physical attributes were evaluated, in particular elevation, slope, aspect, water source, and forest cover were used [36,37]. The digital elevation model with 30 m resolution was downloaded from the science database of the Chinese Academy of Science (http://www.gscloud.cn/, accessed on 1 June 2022). Based on these data, elevation, slope, aspect, water source, and other variables were extracted. Landsat TM (1st survey data from Landsat 5 and the 2nd from Landsat 8) supervised classifications were employed to determine forest cover imagery at 30 m resolution in which different non-forest (value 0) and forest cover (value 1) could be discerned. Water sources were obtained by extracting the valley line by DEM and spatial analysis was conducted in ArcGIS 10.4. The Euclidean distance calculation method accepted the raster layer of distance to water sources. For a detailed method description, see [52].

#### 2.3.2. Human Disturbance Factors

The Euclidean distance calculation method was applied to establish minimum distances between human disturbance factors (such as scenic spots, tourist tracks, residential areas, road or building construction, and farming, with a 30 m × 30 m grid). The residential areas and road data are available from 1:1 million geographical downloads from the National Geomatics Center of China (http://atgcc.sbsm.gov.cn, accessed on 1 June 2022). Huangshan Mountain Management Committee and Gardening Bureau provided data on scenic spots, roads, and building construction (http://hsgwh.huangshan.gov.cn/, accessed on 1 June 2022). Farming data were mainly obtained from our recent line transects. The raster data coordinates of all environmental factors are unified to WGS. 1984 UTM Zone 50N in ArcGIS 10.4.1 with cell size in 30 m × 30 m and ASCII format.

#### 2.3.3. Habitat Suitability Change

We selected macaque occurrence data from the first and second field surveys, environmental variables, and human disturbance factors on Tibetan macaque habitat models to map the macaque habitat suitability change. We divided the occurrence data into ten groups; 75% of the species distribution points were selected as training values, while the remaining 25% were used as test values [53,54]. The area value (AUC) under the receiver operating characteristic curve ROC was used to evaluate the accuracy of predictions. The relative importance of each variable was evaluated using the Jackknife test method. The potential distribution map values from 0 to 1, were divided into four categories: high potential (>0.6), medium potential (0.4–0.6), low potential (0.2–0.4) and non-potential (<0.2) [16,55].

#### 2.3.4. Multilevel Spatial Utilization and Habitat Suitability Change

Multilevel spatial utilization on habitat suitability change was examined using Kernel density estimation (KDE). KDE is a widely used method to calculate animal home range and spatial utilization [55,56].

## 3. Results

### 3.1. Population Distribution and Density

A total of nine groups comprising 327 to 397 Tibetan macaques were recorded. The density was 2.04–2.48 individuals/km^2^, with 20–52 individuals per group and a home range of 2.06–14.13 km^2^. The individuals were located at elevations ranging from 300 to 1700 m. Among them are HX and JLP groups that had larger home range sizes, 14.13 km^2^ and 8.81 km^2^, respectively. However, the home range size of TLG (2.06 km^2^), YAI. (2.44 km^2^), and YAII (3.09 km^2^) groups were smaller (Table 1). They were mainly distributed near the scenic spots in the core touristic areas of Jiulongpo, Tanglingguan, and Hougu. There was a certain degree of overlap and alternation in the home range of different groups (Figure 1).

### 3.2. Population Dynamics of the Yulinkeng I Group

In December 1987, there were 33 individuals in the Yulinkeng group. In December 2020, the group increased to 53 individuals, with an average population size of 38.1 across the survey. When the population size exceeded 50 individuals, the groups tended to fissioned into two. The larger group remained in the original group’s study area. We recorded four such fission events during the study period (Figure 3B). Except for the negative growth of the population in 1988, 1993, 1996, 2005, 2007, 2010, 2014, 2016, and 2020, the other years showed a positive growth rate. The average annual growth rate over the 32 years was 4.15%, the birth rate was 60.82%, the mortality rate was 8.46%, the immigration rate was 4.00%, and the emigration rate was 5.07%; even sex ratios (0.91 ± 0.05), with an average of 8.52 (SE = 0.67) for adult males to 9.35 (SE = 0.51) for adult females, indicated that this is a growing group (Figure 3A).

### 3.3. Human Population, Number of Tourists, Eco-Tourism Income

Population growth in the HMBR was slow, increasing by only 13,268 between 1985 and 2019 (Figure 4A). The number of tourists increased slowly from 1980 to 2004 (from 104,000 to 1,601,900). After being designated a Global Geopark in 2004, the number of tourists increased quickly from 1,601,900 to 3,500,000 (an increase of 118.49%). This increase in tourists has brought in many gate receipts over the past 30 years. Annual eco-tourism income rose from 1,952,000 in 1980 to 3018 million Chinese Yuan in 2020 (Figure 4B).

### 3.4. Predictor Variable Changes and Their Contributions

During the time between the two surveys, environmental variables and human disturbance factors involved in the model prediction underwent apparent changes. Forest cover increased by 22% (67% to 89%) and the number of scenic spots increased from 21 to 65. Road construction increased from 81.2 km to 107.9 km. In contrast, the occurrence of farming decreased from 97 farms to 44 farms. In addition, 17 roads and 21 buildings were repaired during this period (Figure 5).

Table 2 shows that the contribution of environmental variables decreased from 44.21% to 21.32% between the first (1973–1987) and second (2020–2021) survey. The contribution for elevation (from 20.60% to 2.05%), forests cover (from 17.14% to 3.25%), and slope (from 1.55% to 1.29%) also decreased. On the other hand, the contribution of the slope, aspect, and water sources increased (Table 2). However, we found that the human disturbance factors increased from 55.79% to 78.68%. This was mainly due to the substantially increased contributions of scenic spots from 0.71% to 32.88%. The residential area also increased from 2.90% to 14.46%, and farming from 0.67% to 9.97%. On the other hand, road and building construction decreased. It should be noted that, although the contribution of roads was reduced in the second survey, it was still relatively high (14.80%, Table 2).

The MaxEnt model analysis showed that Tibetan macaques had significantly different responses to different factors (Figure 6). Comparing the same variable of different response curves in the first and second field surveys, we found significant differences in the response of Tibetan macaques. For example, regarding environmental variables, in the second survey, the probability of Tibetan macaque presence in low-altitude areas was higher than in the first survey. When comparing the disturbance factors, Tibetan macaques tended to stay away from human-disturbed regions (Figure 6). This can be confirmed by the increase in distance to residential areas (1200 m and 200 m), roads (500 m and 5500 m), scenic spots (500 m and 7000 m), road construction (500 m and 8500 m), and building construction (1500 m and 8500 m) between the first and second surveys (Figure 6).

### 3.5. The Dynamics of Tibetan Macaque Habitat Suitability and Space Use

The accuracy of the Tibetan macaque habitat suitability model during the first (average AUC = 0.816) and second (average AUC = 0.731) surveys were considered “good”. According to the model prediction of two field surveys, the habitat suitability has significantly increased (Figure 7) as follows: good potential (8.18 km^2^ to 12.01 km^2^), medium potential (15.45 km^2^ to 21.82 km^2^), and least potential (113.06 km^2^ to 79.46 km^2^).

Figure 8 shows that the total range area from KDE increased from the 1st to the 2nd survey. Between the two surveys, the profile of each space increased. For example, 25% utilization increased from 3.91 km^2^ to 5.91 km^2^, 75% utilization increased from 16.45 km^2^ to 20.07 km^2^, and 95% utilization increased from 21.94 km^2^ to 26.54 km^2^.

## 4. Discussion

In this study, we found that the population size of Tibetan macaques has significantly increased over the 30-year period surveyed. The 9 groups increased in size from 294 individuals to somewhere between 327 and 397 individuals. Additionally, the macaque population growth rate (11.22–35.03%) was higher than the average (4.15%) of the long-term study group (YAI). This may be related to the emigration of surrounding groups. We found that the monkeys had a higher emigration rate (5.07%) than the long-term study group (YAI). We confirmed that there were at least five groups in the marginal areas of Huangshan (unpublished data). Moreover, in the early 1980s, there was severe hunting, deforestation, and illegal trade in monkeys as laboratory animals, which led to a sharp decrease in the population of Tibetan macaques. In those times, there were not more than 10 individuals in any one group, but the latest survey showed more than 30 individuals were present in most groups [44]. In the 1990s, many prohibitions and forest restoration policies mitigated this situation. Scientific management by local governments restored the number of Tibetan macaques in the area. Furthermore, since 1980, the arrival of more tourists and more ticket income has offered more livelihoods and improved living standards for residents [38]. It also provided more animal protection funds in the respective management departments, easing the tension between humans and the environment. This seems to have been one of the factors influencing the increase in population size of Tibetan macaques. In addition, previous studies suggested that Tibetan macaques can migrate over large areas and need an extensive home range to thrive (more than 7 km^2^ for one group) [43,48]. Nine groups were found in the same area in both surveys, which also shows the accuracy of the two surveys.

This study also found that the distribution of Tibetan macaques expanded from higher to lower elevations (400–800 m). These populations tended to stay away from serious eco-tourism areas in the core zone. Many studies have shown that environmental variables such as elevation, slope, aspect, and forest cover significantly affect animal distribution [57,58,59,60]. This study also supports those findings. However, only two environmental variables, elevation and forest cover, contributed considerably to the Tibetan macaques’ habitat suitability because their behavioral flexibility and overall ecological adaptability are similar to other primates [61]. Among human disturbance factors, residential areas, an extension of scenic road networks, scenic spots, and road and building construction were found to influence Tibetan macaque habitat suitability significantly. All of this resulted in the apparent absence of human disturbance factors. These are beneficial to the conservation of this primate population. Previous studies have shown that Tibetan macaques tend to be present more in the mid to low-elevation areas, because these elevations provide a greater variety of suitable food plants required by the species, compared to higher less vegetatively variable habitats [42,62,63]. Habitat suitability increased at low altitudes and allowed macaques to move farther away from areas of human disturbance, avoiding eco-tourism sites. The methods developed in this study allowed for a more efficient monitoring system to follow the dynamics of the larger Tibetan macaque population. These methods allowed for more efficient and effective protection of the species. In addition, the food species available at low and middle altitudes may also have provided sufficient resources to keep Tibetan macaques in search of food out of core ecotourism interference areas. The higher contribution rate of forest cover can also indirectly confirm this because Tibetan macaques mainly feed on plants. However, whether the distribution change is affected by ecotourism interference or a lack of food resources still needs to be further investigated.

The second survey model showed that scenic spots were the most serious source of eco-tourism disturbance to Tibetan macaque habitats. These scenic spots gradually became the main attraction within this MAB reserve for carrying out eco-tourism. The income of eco-tourism mainly comes from tourist ticket revenue [9,17]. It is generally believed that scenic spots can attract many tourists and provide more human food sources for wildlife, thus changing an animal’s foraging behavior, social interactions, and ranging patterns [17,64]. Moreover, tourist disturbances can increase tension in wilderness areas leading to more attacks. Long-term contact with humans can also lead to an increased risk of infective disease transmission between humans and animals [64,65]. However, in this study, the contribution of scenic spots to the distribution of Tibetan macaques was high. The increase in suitable habitat areas at low altitudes undoubtedly reduced the impact of tourists on the macaques’ lives because more tourist activities were concentrated in the core scenic spots.

The success of the HMAB reserve for the sustainable development of eco-tourism and protecting the Tibetan macaque is apparent due to the increased eco-tourism revenues, Tibetan macaque population growth, and the increase in habitat suitability. However, we still believe that eco-tourism disturbance in the Tibetan macaque habitat needs further control. Further research on the scale of disturbance by this activity is required. This can be conducted by reducing the influence of high interference factors such as farming, road, scenic spot, and building construction. Management offices in Huangshan have already started these efforts. For example, they are carrying out a series of targeted poverty alleviation policies, implementing strategies that will help increase residents’ income. For instance, growing monetary subsidies to reduce agricultural activities and increase related technical training. Regarding the impact of roads and further road construction on the environment, they are decreasing the construction of new roads and using environmentally friendly materials when constructing new ones. It is worth recalling that since the beginning of 1987, the management has adopted a policy of regularly closing down scenic spots at intervals of 3–5 years to allow them to regenerate. In addition, as far as meeting the basic needs of sightseeing time, moving the location of the ropeway downwards will encourage tourists to walk around Huangshan more. We also need to pay attention to community development and reduce deforestation by providing suitable alternative energy sources to wood, such as hydroelectricity and natural gas. Increased linkages with external areas through sustainable eco-tourism will also help. Efforts should also be made to increase the control of buffer zone areas through better land use management, natural resource protection, infrastructure construction, and scenic spot development. These methods could reduce many eco-tourism related disturbances on the Tibetan macaque habitat.

## 5. Conclusions

The purpose of the MAB program is to systematically clarify the structure and function of different regions of MAB reserves and to predict the changes in the biosphere and its resources caused by human activities. It also strives to predict the impact of this change on human beings using the tools of natural science, the social sciences, humanities and applied technology. The work is carried out by scientists, technicians, manufacturing management staff, political policymakers, and the local community. This is done to responsibly utilize and protect the biosphere’s resources, preserve genetic diversity, and improve the relationship between humans and the environment. Eco-tourism is widely used as a sustainable development measure for constructing MAB reserves. However, whether it can promote the goal of sustainable development in these reserves, improve the economy and protect animal protections remains to be verified. In this study, we aimed to evaluate the effectiveness of the sustainable development goals in this MAB reserve from the perspective of population ecology and the biological conservation of Tibetan macaques as a model species. Specifically, we expected to establish an evaluation system for the sustainable development goals based on the long-term field survey data on eco-tourism and Tibetan macaque population dynamics in Huangshan. The MaxEnt model was used to predict changes in the biosphere and resources caused by human disturbance.

This study found that the economy of the HMAB reserve has been developing rapidly, with a significant rise in residents’ income over the past thirty years. The population and habitat suitability of Tibetan macaques increased. They also showed a trend of habitat expansion to lower altitude areas where eco-tourism disturbance was low. The results confirm that the development of eco-tourism in the HMAB reserve can promote sustainable development of the economy and help conserve wildlife populations. Nevertheless, further changes in the distribution pattern of the Tibetan macaque population from the core area (high altitude) to the buffer zone (low altitude area) still requires close attention. The buffer zone faces more interference from residential areas, such as roads, vehicles, lights, and artificial food. Monkeys will appear in these artificial environments if measures are not properly taken. Due to the implementation of the ecological restoration policy, a large area of plantation forest, bamboo forest, and abandoned farmland has been reclaimed. Therefore, the importance of buffer zones should be enhanced, although this may violate previous environmental protection concepts of buffer zone protection. It is likely to become a new problem faced by the MAB. reserve with the continuation and expansion of eco-tourism. Since eco-tourism involves a wide range of political, economic, cultural, and demographic issues, our findings can only indicate the impact of eco-tourism on the population ecology of an endemic endangered primate. In the next phase of this study, continued monitoring of macaques in the buffer zone is needed. This will help to better determine the drivers and stressors of eco-tourism on their population ecology and to further reveal the sustainable development effectiveness in this MAB reserve.

In summary, a clear trend of sustainable economic development increase, expansion of the Tibetan macaque population, and an improved natural environment in Huangshan under the MAB Program was documented.

## Figures and Tables

**Figure 1 biology-11-01042-f001:**
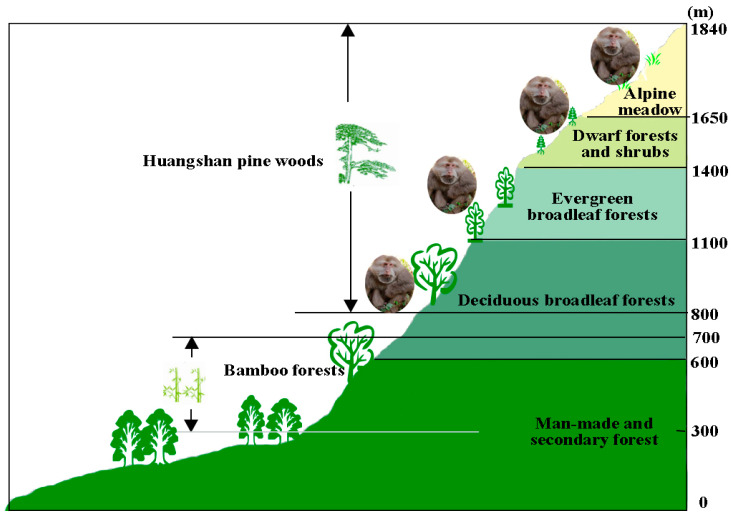
The vertical distribution of vegetation in HMBR. Figure modified from [38].

**Figure 2 biology-11-01042-f002:**
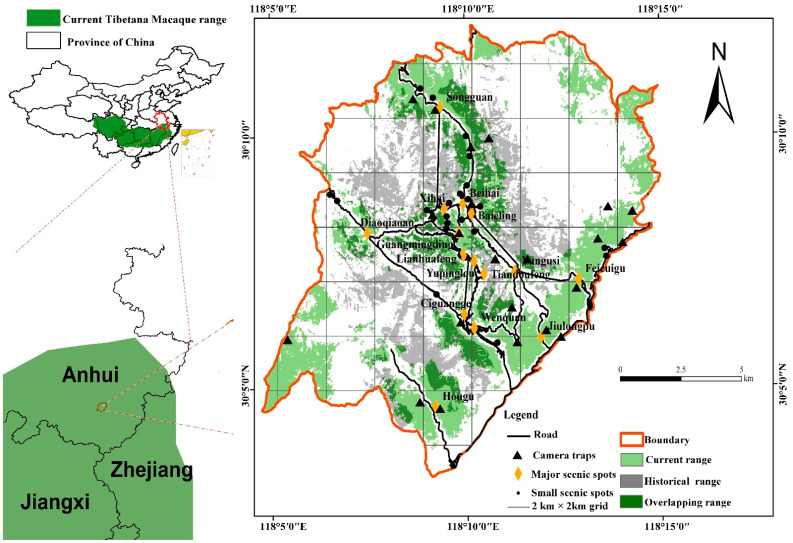
The Huangshan study area (HMBR), Anhui Province, China.

**Figure 3 biology-11-01042-f003:**
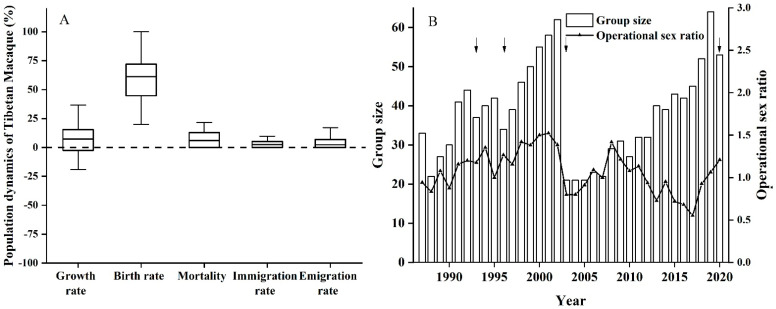
Population dynamics of Tibetan macaques. (**A**) the annual net growth rate, birth rate, mortality, immigration, and emigration rate. (**B**) the group size and operational sex ratio in the Yulingken A1 group, with the group fission events indicated by the arrow.

**Figure 4 biology-11-01042-f004:**
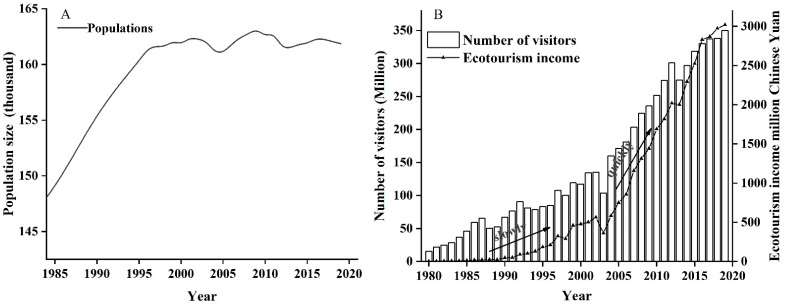
The local population, visitors, and annual eco-tourism income in HMBR. (**A**) local people in Huangshan district from 1985 to 2020 (Data source: *Huangshan City Statistical Yearbook*). (**B**) shows the number of visitors and eco-tourism incomes in HMBR from 1980 to 2019 (Data source: *Huangshan Scenic Area Management Committee*).

**Figure 5 biology-11-01042-f005:**
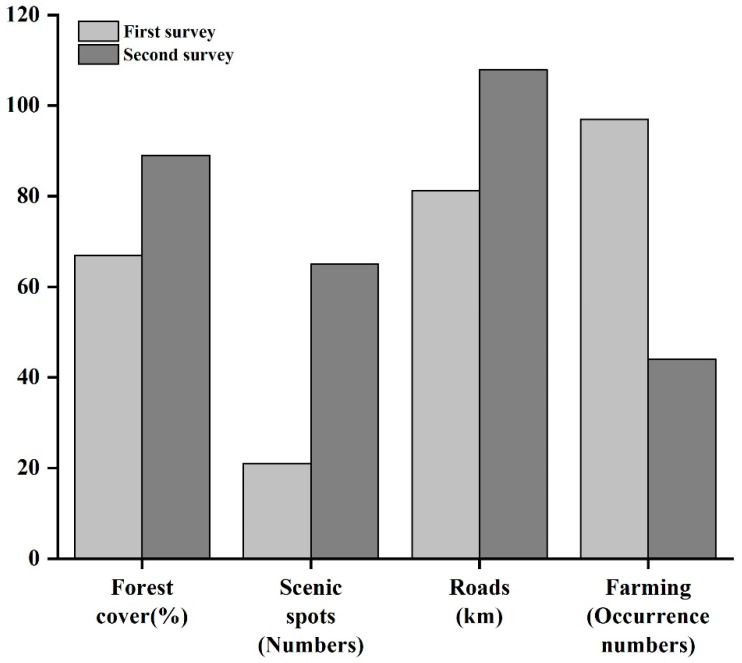
Predictor variable changes in HMBR from the first (1973–1987) and second (2020–2021) surveys.

**Figure 6 biology-11-01042-f006:**
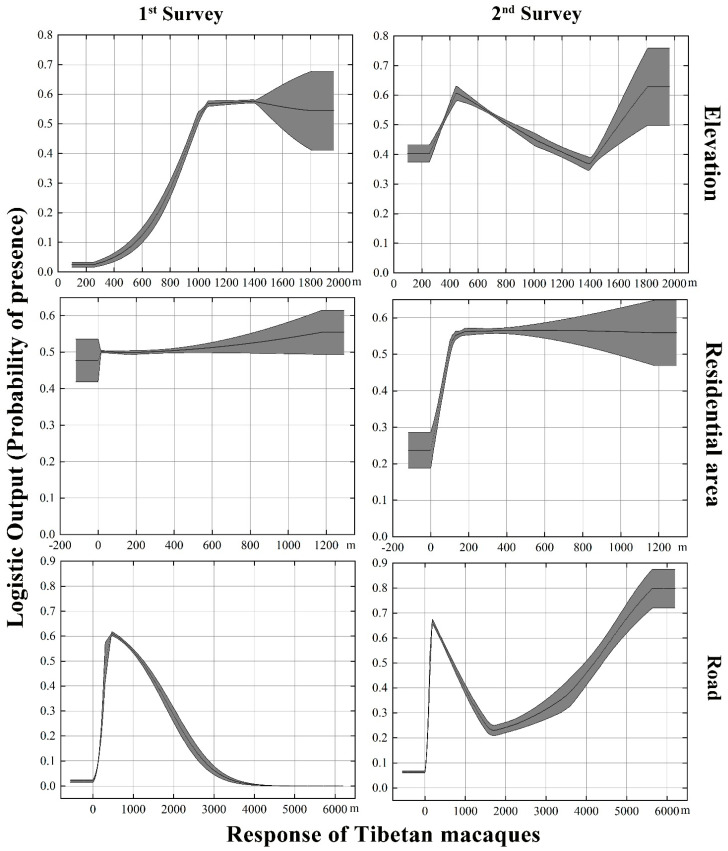
Response of Tibetan macaques to the most important factors affecting habitat suitability in the first (1973–1987) and second (2020–2021) surveys.

**Figure 7 biology-11-01042-f007:**
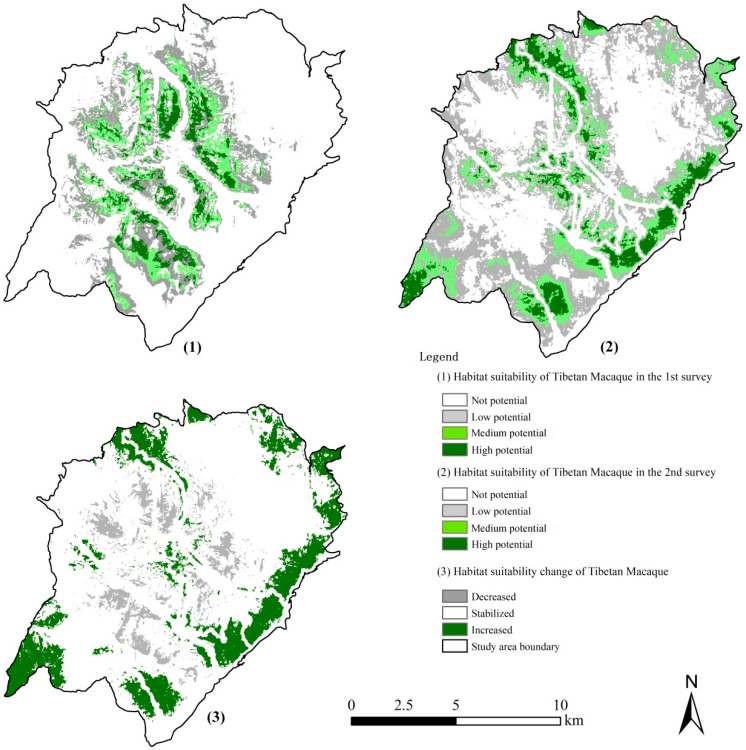
Tibetan macaque habitat suitability changed in HMBR between the first (1973–1987) and second (2020–2021) surveys.

**Figure 8 biology-11-01042-f008:**
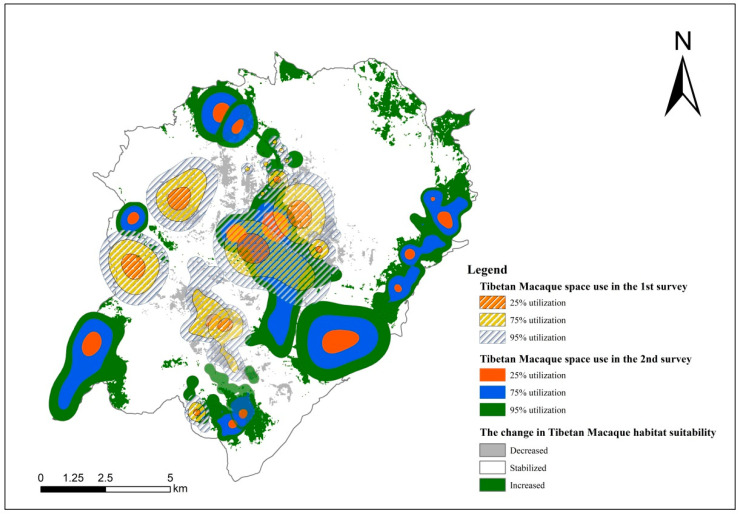
Change in Tibetan macaque space use in HMBR from the first (1973–1987) to the second (2020–2021) surveys.

**Table 1 biology-11-01042-t001:** The population size, density, and distribution of Tibetan macaques in HMBR. The home range sizes were determined by KDE.

Group Name	Group Size	Range Size (km^2^)	Range/Altitude (m)	Data Sources
YAI	52	2.44	600–1200	Long-term study
YAII	40–50	3.09	600–1300	The long-term study, Line transects, Camera traps
TBSII	40–50	7.07	500–1300	Line transects, Camera traps
JLP	40–50	8.81	400–1100	Line transects, Camera traps
TLG	30–40	2.06	700–1600	Line transects, Camera traps
FRL	40–50	3.63	500–1500	Line transects, Camera traps
SG	20–30	3.44	500–1600	Line transects, Camera traps
HX	30–40	14.13	1100–1700	Line transects, Camera traps
THII	35	7.13	300–600	Long-term study
Total	327–397	51.8	-	-

**Table 2 biology-11-01042-t002:** The changes in contribution value of different predictor variables in Tibetan macaque habitat suitability between the first (1973–1987) and second (2020–2021) surveys.

Predictors Variables	Contribution Value	
	First Survey (1973–1987)	Second Survey (2020–2021)
Elevation	20.60%	2.05%
Slope	1.55%	3.30%
Aspect	1.67%	3.92%
Water sources	3.25%	8.80%
Forest cover	17.14%	3.25%
Farming	0.67%	9.97%
Residential area	2.90%	14.46%
Road	19.23%	14.80%
Scenic spots	0.71%	32.88%
Road construction	18.91%	0.59%
Building construction	13.36%	5.97%

## Data Availability

The data presented in this study are available in the manuscript.

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
