# Peer review of "Ecotourism Disturbance on an Endemic Endangered Primate in the Huangshan Man and the Biosphere Reserve of China: A Way to Move Forward"

_biology, 2022, doi:10.3390/biology11071042_

Round 1

Reviewer 1 Report

The article illustrates the results of a long-term study conducted on the Tibetan macaque and its relationship with humans in the Huangshan Man and Biosphere Reserve. This is a very important topic and the paper can be a good example of the kind of interaction between human beings and macaques and impact of ecotourism in this type of protected area. The authors present a comparison between data obtained from a long-term study with recent data.

Comments and suggestions:

a) The authors should revise English especially in the abstracts and discussion. Although not a native English speaker even for me it is easy to see that there are some problems: 

 the concepts are sometimes exposed not in a continuous form, but interrupted with wrong punctuation. Occasionally there are unclear or unfinished sentences.  I recommend a revision.
E.g. see lines  23, 73, 74

b) Line 20: if you are referring to the MAB Unesco project it is correct to use the word man but you must specify it because the word “man” it is now obsolete and it is better to use the word humans. 

c) Results (and methods)
it is not clear which are the two surveys (first and second) mentioned in the results. 

Which periods are being compared?

You mentioned long term data (30 years): are you comparing this data, but which period? Not clear.
First survey: is this the one conducted in 1987? If yes, to what extent it is possible to compare these two surveys conducted in different time and, I think with different techniques? Please specify:  what is the first survey, when and how long it was and why it is possible to compare it with the second survey. The work you mention (47) is in Chines and also the website you mentioned for methods. You could give a summary or synopsis of the methods and results of this first survey (if this is the first!).

Second survey: I assume is the one done from March 2020 until March 2021 but here you should specify the survey days, if it is the whole month, years, hours of observation etc. Not clear. 

In sum: the paper does not clearly present the methods of data collection - it refers to previous works in Chinese: a paper and a web page - and show the results of the first and second surveys without mentioning them in a clear way in the methodology.

I think this is the biggest problem with the paper and I urge the author to make this part clearer.

Author Response

The article illustrates the results of a long-term study conducted on the Tibetan macaque and its relationship with humans in the Huangshan Man and Biosphere Reserve. This is a very important topic and the paper can be a good example of the kind of interaction between human beings and macaques and impact of ecotourism in this type of protected area. The authors present a comparison between data obtained from a long-term study with recent data.

Comments and suggestions:

  1. a) The authors should revise English especially in the abstracts and discussion. Although not a native English speaker even for me it is easy to see that there are some problems: the concepts are sometimes exposed not in a continuous form, but interrupted with wrong punctuation. Occasionally there are unclear or unfinished sentences. I recommend a revision.

E.g. see lines  23, 73, 74

Answer: We thank you for your comments. We agree with the reviewer. We invited a Native English speaker to modify the language of the manuscript. And We have made significant changes to our manuscript based on your critique and feel that we have strengthened our manuscript.

  1. b) Line 20: if you are referring to the MAB Unesco project it is correct to use the word man but you must specify it because the word “man” it is now obsolete and it is better to use the word humans.

Answer: We thank you for your comments. We have changed all the word “man” to specify “Man”

  1. c) Results (and methods)

it is not clear which are the two surveys (first and second) mentioned in the results.

Which periods are being compared?

You mentioned long term data (30 years): are you comparing this data, but which period? Not clear.

First survey: is this the one conducted in 1987? If yes, to what extent it is possible to compare these two surveys conducted in different time and, I think with different techniques? Please specify:  what is the first survey, when and how long it was and why it is possible to compare it with the second survey. The work you mention (47) is in Chines and also the website you mentioned for methods. You could give a summary or synopsis of the methods and results of this first survey (if this is the first!).

Second survey: I assume is the one done from March 2020 until March 2021 but here you should specify the survey days, if it is the whole month, years, hours of observation etc. Not clear.

In sum: the paper does not clearly present the methods of data collection - it refers to previous works in Chinese: a paper and a web page - and show the results of the first and second surveys without mentioning them in a clear way in the methodology.

I think this is the biggest problem with the paper and I urge the author to make this part clearer.

Answer:We thank you for your comments. We have significantly revised the methods and results in the revised version.

Reviewer 2 Report

General comments:

This was a very comprehensive and detailed study and I can definitely feel the passion and the expertise of the authors. I am certain this will provide great and beneficial information for the reserve. Thank you.

In general, I would like more explanations throughout the text and I would like to be helped more as a reader. Please write everything instead of just referring to a reference. Please also use years instead of survey 1 and 2. The discussion holds much information I would have liked to have known in the introduction. Perhaps make the introduction more specific to the area.

I think it would benefit the text to explain more throughout and mention what is referred to. For example line 199, it just says to see ref 47. Remember, that you want to help the reader read and understand your article, and if you have to constantly check references you lose the reading flow.

Please insert missing words throughout the text, such as “is” “are” “were”. Thank you

You may need to reconsider if you are indeed talking about ecotourism, or just nature tourism.

Keywords:

Suggest to change the words already mentioned in the title to other keywords to increase chance of occurring in searches

Introduction:

The introduction contains much relevant information on the topic discussed here, including both pros and cons of ecotourism and development. However, it would improve the introduction to have a more detailed outline of what the Huangshan Reserve wishes to achieve. There is much discussion about conservation, but what is conservation to the reserve? What are the goals of the management?

The introduction becomes very general, but it would be nice for it to be more specific for the reader to be able to understand the Reserve and the challenges the reserve faces. For example, you could include more specific problems found in the reserve. – and a historical cultural overview.

Line 113: add “as”; serve as

Line 142-145: What are the exact goals for the reserve? – wish to protect how much? To earn how much? Etc…

Line 151-160: slightly confusing, should be rewritten starting with all historical information and ending with current info

Methods:

Many different variables are included in the methods, which is good. It seems very thorough and as the authors have included everything possible.

Line 193: attach should be attack?

Line 199: write what ref 47 is

Line: 238: Please briefly explain the age and sex classes instead of again expecting the reader to read other articles to understand yours.

Line 297-300: Which GPS coordinates were used? KDE is parametric, so wondering if that was taken into consideration with the GPS positions. – and apologies if this was explained elsewhere.

Results:

Line 328: I am not able to see the colours (red arrow e.g.)

Line 337: Gate receipts? Do you mean they bought many entrance tickets and thereby an increase in income?

Line 352: I cannot seem  to remember when the first and the second survey were? Can you somewhere underline when which survey was? I know you have data dating back to 1985, but it has become blurry when what happened. Possibly because you have many variables and many different methodologies. Perhaps instead of writing first and second survey, write the years in the table and in the text.

Line 364-395: I believe you may need to be a bit more wary of the words used in this section. They habitat suitability analysis does not show if the habitat is suitable for the macaques. It shows which habitats the macaques prefer. You are showing habitat preference, not habitat suitability. I know this is the name of the analysis but it is not what it means*. The habitat preference of the macaques has increased in general because they have adapted to the changing environments over time. For example: They have adapted to human food sources, so when the availability of human food sources increase you will see an increase in preference for those areas.

*Perhaps refer to Hansen et al., 2021 on habitat preference in long-tailed macaques.

You mention that forest cover has increased but which forest cover? An increase in fruiting trees that they forage on or an increase in other trees? How did the increase happen? – active reforestation?

Discussion:

Line 431-435: I do not understand these sentences. Please elaborate

Line 442-443: Please explain these goals in introduction for the reader to understand

This discussion takes the reader more on a journey and explains much more about the reserve making it more personal and enjoyable to read. I would have liked to have known some of this in the introduction. – and even more that could lead to a better understanding here.

You may need to dig a bit deeper into your results to see what they actually mean. Habitat suitability is a great tool but it contains a lot of information that should be addressed more than just saying that suitability increased, because it may not have. Preference may have, but why? If you coose to keep all of your response variable curves, these should also be discussed better. You could also move them to the appendix. Although they do hold a lot more information than addressed here.

I appreciate that you say your results are indicative only and that more research is needed. I think you are right and it would be great to know more about the reserve. At the same time I feel you have more information in for example the habitat suitability analysis that you could tap into.

Author Response

This was a very comprehensive and detailed study and I can definitely feel the passion and the expertise of the authors. I am certain this will provide great and beneficial information for the reserve. Thank you.

Answer: We thank you for your comments.

In general, I would like more explanations throughout the text and I would like to be helped more as a reader. Please write everything instead of just referring to a reference. Please also use years instead of survey 1 and 2. The discussion holds much information I would have liked to have known in the introduction. Perhaps make the introduction more specific to the area.

Answer: We thank you for your comments. We have made significant changes to our manuscript based on your critique and feel that we have strengthened our manuscript.

I think it would benefit the text to explain more throughout and mention what is referred to. For example line 199, it just says to see ref 47. Remember, that you want to help the reader read and understand your article, and if you have to constantly check references you lose the reading flow.

Answer: We thank you for your comments. We have made changes in the revised manuscript according to your suggestions.

Please insert missing words throughout the text, such as “is” “are” “were”. Thank you

Answer: We thank you for your comments. We have made changes in the revised manuscript according to your suggestions.

You may need to reconsider if you are indeed talking about ecotourism, or just nature tourism.

Answer: We thank you for your comments. We are indeed talking about ecotourism, We are sorry for your misunderstanding. We have revised it in the Introduction and Discussion.

Keywords:

Suggest to change the words already mentioned in the title to other keywords to increase chance of occurring in searches

Answer: We thank you for your comments. We have changed the keywords. The new keywords are Man and the Biosphere Reserve; Sustainable development; Ecotourism; Tibetan macaque; Species distribution modeling; MaxEnt

Introduction:

The introduction contains much relevant information on the topic discussed here, including both pros and cons of ecotourism and development. However, it would improve the introduction to have a more detailed outline of what the Huangshan Reserve wishes to achieve. There is much discussion about conservation, but what is conservation to the reserve? What are the goals of the management?

The introduction becomes very general, but it would be nice for it to be more specific for the reader to be able to understand the Reserve and the challenges the reserve faces. For example, you could include more specific problems found in the reserve. – and a historical cultural overview.

Answer: We thank you for your comments. We have made changes in the revised manuscript according to your suggestions.

Line 113: add “as”; serve as

Answer: We thank you for your comments. We have added “as” in the revised manuscript.

Line 142-145: What are the exact goals for the reserve? – wish to protect how much? To earn how much? Etc…

Answer: We thank you for your comments. We have added “However, in recent years, with the increase of ecotourism tourists, more roads, scenic spots, and buildings are constantly being built and renovated. Whether these have caused damage to the ecosystem of HMAB has not been studied. In addition, there is a lack of practical data to support whether the food residues brought by more tourists and the food fed impact animal populations and distribution. Finally, the economic growth dividends brought by eco-tourism have increased the income of local farmers and reduced their de-pendence on farmland. Most of the cultivated land has now become forest land. Will this have a positive impact on the distribution and population of animals? Therefore achiev-ing the collaborative and sustainable development of nature conservation and tourism is essential to preserving biological diversity and a real problem for scientific management in Huangshan”.

Line 151-160: slightly confusing, should be rewritten starting with all historical information and ending with current info

Answer: We thank you for your comments. We have added “Since 1983 the behavioral ecology and population dynamics of the Yulin Keng I group (YAI) of wild Thibetan macaques has been continuously studied in the HMAB. In recent years, it has been found that the Tibetan macaques' population (around YAI)has increased significantly. And human-monkey conflicts often occur in HMAB. Therefore, there is an urgent need to carry out special investigations and research on the population and distribution of Tibetan macaques”.

Methods:

Many different variables are included in the methods, which is good. It seems very thorough and as the authors have included everything possible.

Line 193: attach should be attack?

Answer: We thank you for your comments. We have changed “attach” to “attack”.

Line 199: write what ref 47 is

Answer: We thank you for your comments. We have added some information about the ref 47.

Line: 238: Please briefly explain the age and sex classes instead of again expecting the reader to read other articles to understand yours.

Answer: We thank you for your comments. We have added “The age-sex classes were determined based on differences in body size, hair color, facial color, reproductive organs, and behavior.”

Line 297-300: Which GPS coordinates were used? KDE is parametric, so wondering if that was taken into consideration with the GPS positions. – and apologies if this was explained elsewhere.

Answer: We thank you for your comments. we have explained in line294 The raster data coordinates of all environmental factors are unified to W.G.S. 1984 UTM Zone 50N in ArcGIS 10.4.1 with cell size in 30 m×30 m and ASCII format.

Results:

Line 328: I am not able to see the colours (red arrow e.g.)

Answer: We are so sorry for this mistake and thank you for your comments. We have deleted the “Red” in the revised manuscript.

Line 337: Gate receipts? Do you mean they bought many entrance tickets and thereby an increase in income?

Answer: We thank you for your comments. The main income of the HMBR is their tickets.

Line 352: I cannot seem  to remember when the first and the second survey were? Can you somewhere underline when which survey was? I know you have data dating back to 1985, but it has become blurry when what happened. Possibly because you have many variables and many different methodologies. Perhaps instead of writing first and second survey, write the years in the table and in the text.

Answer: We thank you for your comments. We have written the years in the revised manuscript.

Line 364-395: I believe you may need to be a bit more wary of the words used in this section. They habitat suitability analysis does not show if the habitat is suitable for the macaques. It shows which habitats the macaques prefer. You are showing habitat preference, not habitat suitability. I know this is the name of the analysis but it is not what it means*. The habitat preference of the macaques has increased in general because they have adapted to the changing environments over time. For example: They have adapted to human food sources, so when the availability of human food sources increase you will see an increase in preference for those areas.

*Perhaps refer to Hansen et al., 2021 on habitat preference in long-tailed macaques.

You mention that forest cover has increased but which forest cover? An increase in fruiting trees that they forage on or an increase in other trees? How did the increase happen? – active reforestation?

Answer: Thank you for your valuable comments, and we agree with you. In the manuscript, we did make a mistake in our description. MaxEnt model can predict their potential habitat, which does not mean that it is their suitable habitat, but only the area of possible preference.Thank you for mentioning the influencing factor of forest cover, which is a very interesting topic. Only because in this study, we only pay attention to the large-scale changes of forest cover in the region for more than 30 years and do not consider what type of vegetation, or whether it is the growth of their feeding plants because we have not conducted a systematic study on the feeding plants of Tibetan monkeys, In the follow-up work, we want to further study their microenvironment.

Discussion:

Line 431-435: I do not understand these sentences. Please elaborate

Answer: Thank you for your comments. We have rewritten these hard-to-read sentences.

Line 442-443: Please explain these goals in introduction for the reader to understand

Answer: Thank you for your comments. We have made changes in the revised manuscript according to your suggestions.

This discussion takes the reader more on a journey and explains much more about the reserve making it more personal and enjoyable to read. I would have liked to have known some of this in the introduction. – and even more that could lead to a better understanding here.

You may need to dig a bit deeper into your results to see what they actually mean. Habitat suitability is a great tool but it contains a lot of information that should be addressed more than just saying that suitability increased, because it may not have. Preference may have, but why? If you coose to keep all of your response variable curves, these should also be discussed better. You could also move them to the appendix. Although they do hold a lot more information than addressed here.

I appreciate that you say your results are indicative only and that more research is needed. I think you are right and it would be great to know more about the reserve. At the same time I feel you have more information in for example the habitat suitability analysis that you could tap into.

Answer: Thank you for your comments. We have made changes in the revised manuscript according to your suggestions.

Reviewer 3 Report

It is a nice work on assessing disturbance caused by ecotourism activities toward wildlife (here is the Tibetan macaques). However, to make the reader understand on the aims and objectives of the paper, the paper needs to:

- modify the title

-  re-structure the introduction

- emphasize on strengthen the connection between sustainable development - economic changes - dynamics of human - and disturbance (still a bit loose) as this is the key point of and provide sufficient justification on why only for a single spesies

- minor typos and unfinished/unclear sentences

- more details explanation on data sources in particular differences in data input

- in depth thinking on the results so then it will be easier to compile into a comprehensive discussion that fulfilled the aims of the study.

Thank you very much.

Author Response

It is a nice work on assessing disturbance caused by ecotourism activities toward wildlife (here is the Tibetan macaques). However, to make the reader understand on the aims and objectives of the paper, the paper needs to:

modify the title

Answer: We thank you for your suggestions. We have revised the text.

-  re-structure the introduction

Answer: We thank you for your comments. We have significantly revised the introduction in the revised version.

- emphasize on strengthen the connection between sustainable development - economic changes - dynamics of human - and disturbance (still a bit loose) as this is the key point of and provide sufficient justification on why only for a single spesies

Answer: We thank you for your comments. We have significantly revised the introduction and Discussions in the revised version.

- minor typos and unfinished/unclear sentences

Answer: We thank you for your comments. revised the minor typos and unfinished/unclear sentences in the revised version, according to your suggestions.

- more details explanation on data sources in particular differences in data input

Answer: We thank you for your comments. We have significantly revised according to your suggestions.

- in depth thinking on the results so then it will be easier to compile into a comprehensive discussion that fulfilled the aims of the study.

Answer: We thank you for your comments. We have made major changes to our manuscript based on your critique, and feel that in doing so we have strengthened our manuscript.

Thank you very much.

I suggest to delete the 'sustainable development' from the title as the study focus is on how ecotourism activities might pose disturbances for the endemic endangered primate in HMBR and change it to:

Ecotourism disturbance on an endemic endangered primate in the HMBR of China: A way to move forward

Answer: We thank you for your comments. We think your comments are better.

L19-22 this is a key point of your study but there is a need in re-structuring introduction and also discussion part where the connection between sustainable development - economic changes - dynamics of human - and disturbance a bit loose

Answer: We thank you for your comments. We have revised these.

L38-38 This study explored the sustainable development policy of HMBR based on 30 years data of monitoring the Tibetan macaque, local human population, visitors, and annual ecotourism income in Huangsan by estimating species habitat suitability and the impact of ecotourism

Answer: We thank you for your comments. We have revised these.

L41-42 How low? provide the range

Answer: We thank you for your comments. We have added,They expanded their habitat to the low-altitude buffer zone (400-800m)”.

L42 is this statistically tested? then provide the p value

Answer: We thank you for your comments. (the substantially increased contributions of scenic spots from 0.71% to 32.88%) we have added the revised text.

L57 why in the mountainous areas? this is not clear

Answer: We thank you for your comments. We have added, “where the economy is dependent on natural resources for its growth and development. More agricultural activities such as logging, farming, and grazing will aggravate the de-struction of the natural environment”

L64 what 'sites' here imply to? ecotourism or links?

Answer: We thank you for your comments. We means “The MAB implementing ecotourism”

L72 how could ecotourism activities generate carbon emissions of animals and plants? I think it is better to classify and elaborate on the impacts of ecotourism to physical attributes, ecological attributes which also include the changes of animal behaviour, stress, disturb feeding and foraging pattern, even disturb reproduction, socialization time, etc and also social-cultural attributes. But emphasizing on the negative impacts towards wildlife would be more suitable as this paper discuss about ecotourism Vs Tibetan Macaque. or move the negative impacts of ecotourism to paragraph 4

Answer: We thank you for your comments. We have significantly revised according to your suggestions.

L86-87 this sentence is unclear. first it is mentioned about animals then it changed into important current activities of conservation biology

Answer: We thank you for your comments. We have changed this sentence to “Animals are an essential part of the natural ecosystem. Research and preservation of an-imals' habitat selection, distribution, and demography are some of the more critical cur-rent activities of conservation biology.”

L95 why is single species? provide justification on why, when looking at the impacts on ecotourism this study only focus on a species? is it because the species is in threatened or endangered conditions? or is it a flagship or key stone species?

Check in the discussion, have you related the issue of single spe

Answer: We thank you for your comments. We have significantly revised according to your suggestions.

L101 only in protected areas?some ecotourism activities might occur in non protected areas but important habitats for (a or many) wildlife species

Answer: We thank you for your comments. We mean “Multiple studies have shown that the impact of tourists walking activities can also change the spatial distribution of soil structure and vegetation” we have changed the “protected areas” to “tourists walking activities”

L128 All of these factors pose... to primate conservation and also to regional economic development.

Answer: We thank you for your comments. We have revised it according to your suggestions.

L157 is there any specific activity of what consitute of economic development and on what scale? and for human activities, are these always mean ecotourism activities?

Answer: We thank you for your comments. We have revised it according to your suggestions.

I suggest to re-structure your background starting from:

Ecotourism as a tool in achieving sustainable development as well as   biodiversity conservation plus positive impact of ecotourism.

Ecotourism mostly occurs in conservation areas this included MAB. Then explain about MAB reserve and its roles as well as successful MAB in boosting sustainable development in particular economic aspect with the 3 pillars of ecotourism.

Concerns on negative impact of ecotourism in particular to the management - explain on various impacts of ecotourism include towards wildlife

Long term effect of ecotourism to wildlife still limited - for example primate and why this kind of study is required.

Huangsan, its ecotourism and Tibetan macaque and their problems. provide evidence on how ecotourism activity may generate negativev impact on  Tibetan macaques.

The aims of this study plus the hypotheses if required

Answer: Thank you very much, we agree with your comments and we have revised it according to your suggestions.

L211 any possible/relevant information

Answer: Thank you very much, we have deleted it.

L223:is this for the dominant group only?

Answer:Yes.

L231 please be aware of the differences in the data obtained for different variables

Answer: We thank you for your comments. We have revised it in the text.

L236-238 Unfinished sentence,according to what?

Answer: We thank you for your comments. The population's age-sex class was classified according to Li et al. (2020) The age-sex classes were determined based on differences in body size, hair color, facial color, reproductive organs, and behavior.

L248-251 were these data taken from the current/studied population or other group?

Answer: these data from “The long-term population data for the Yulinkeng I group (a long-term field study of Tibetan macaques)collected over more than 30 years was conducted by the International Collaborative Research Center for Huangshan Biodiversity and Tibetan Macaque Behavioral Ecology" (http://hsmonkey.ahu.edu.cn/)”. The YAI group is also included in this study.

L259-260 This method allowed a larger scale prediction of potential macaque distribution areas that beyond the sampling areas with high predictive accuracy

Answer: We thank you for your comments. We have revised it in the text.

L264-276 To project and map habitat suitability of Tibetan macaques,wide physical attributes were used in particular elevation, slope, aspect, water source, and forest cover were used.

Landsat TM (1st survey data from Landsat 5 and the 2nd from Landsat 8 supervised classification were employed to determine forest cover imagery at 30 m resolution in which differed non-forest (value 0) and forest covers  (value 1).

The Euclidean distance calculation method was applied to establish...

Answer: We thank you for your comments. We have revised it in the text. We think your comments are better. and feel that in doing so we have strengthened our manuscript.

L298-299 and L304

Answer: We thank you for your comments. We have revised it in the text. We think your comments are better. and feel that in doing so we have strengthened our manuscript.

L382 was considered 'good'

Answer: We thank you for your comments. We have revised it in the text.

L425 How about the relationship with food availability in that specific period? have this taken into account?not merely an avoidance behaviour towards ecotourism activities.

besides that, the scale of ecotourism activities needs to be measured as well to prove that the activities might generate negative impacts to the population of Tibetan macaques

Answer: We thank you for your comments. We have revised it in the text. In addition, the food available at low and middle altitudes may also lead the Tibetan macaque to stay away from serious ecotourism interference core areas. The higher contribution rate of forest cover can also indirectly confirm this because Tibetan macaques mainly feed on plants. However, whether the distribution change is affected by ecotourism interference or the lack of food resources still needs further thematic research.

L436 the mid to low elevation may provide bigger chances of various food plants required by the species as low to mid had richer type of plants species compared to the higher ground.

Answer: We thank you for your comments. We have revised it in the text.
